# AWS-DAIE: Incremental Ensemble Short-Term Electricity Load Forecasting Based on Sample Domain Adaptation

**Shengzeng Li** [1,2], **Yiwen Zhong** [1,2] **and Jiaxiang Lin** [1,2,*]

1   College of Computer and Information Sciences, Fujian Agriculture and Forestry University, Fuzhou 350002, China
2   Key Laboratory of Smart Agriculture and Forestry (Fujian Agriculture and Forestry University), Fujian Province University, Fuzhou 350002, China
*   Correspondence: linjx@fafu.edu.cn

**Abstract:** Short-term load forecasting is a prerequisite and basis for power system planning and operation and has received extensive attention from researchers. To address the problem of concept drift caused by changes in the distribution patterns of electricity load data, researchers have proposed regular or quantitative model update strategies to cope with the concept drift; however, this may involve a large number of invalid updates, which not only have limited improvement in model accuracy, but also insufficient model response timeliness to meet the requirements of power systems. Hence, this paper proposes a novel incremental ensemble model based on sample domain adaptation (AWS-DAIE) for adapting concept drift in a timely and accurate manner and solves the problem of inadequate training of the model due to the few concept drift samples. The main idea of AWS-DAIE is to detect concept drift on current electricity load data and train a new base predictor using Tradaboost based on cumulative weighted sampling and then dynamically adjust the weights of the ensemble model according to the performance of the model under current electricity load data. For the purposes of demonstrating the feasibility and effectiveness of the proposed AWS-DAIE algorithm, we present the experimental results of the AWS-DAIE algorithm on electricity load data from four individual households and compared with several other excellent algorithms. The experimental results demonstrated that the proposed AWS-DAIE not only can adapt to the changes of the data distribution faster, but also outperforms all compared models in terms of prediction accuracy and has good practicality.

**Keywords:** short-term electricity load forecasting; concept drift; cumulative weighted sampling; sample domain adaptation

## 1. Introduction

Electricity load forecasting has been attracting research and industry attention due to its important role in the energy management system, which is the basis for energy production and distribution and supply, as well as an important component of intelligent power systems. Accurate short-term electricity load forecasting not only helps to promote energy saving and emission reduction projects in the power system, but also helps to operate the power system reliably and safely. According to the time scale, electricity load forecasting can be divided into ultra-short-term electricity load forecasting, short-term electricity load forecasting, medium-term electricity load forecasting, and long-term electricity load forecasting. Among them, short-term load forecasting predicts the electricity load values for the next few hours or days [1], and the forecast's results are used as a basis for planning the mobilization of power system units. The volatility of the electricity load affects the optimal dispatch of the power system, and with the large-scale grid integration of distributed energy sources [2], the difficulty of forecasting short-term electricity load accurately has increased further due to the dramatic increase in volatility and nonlinearity in the electricity load data.

In recent years, deep learning methods have achieved great success in the field of short-term electricity load forecasting because of their ability to model complex systems [3–5]. However, there is still a large gap in the application of traditional offline deep learning methods to the industry. Traditional offline deep learning models all historical data, that is all data should be available during training. For smart meters [6], only a part of the data is available in the early stage of training, and the data will be delivered continuously, which requires the traditional offline model to retrain the model by combining historical data and newly arrived electricity load data. Obviously, this is not realistic; it requires much wasted computing resource. However, the distribution pattern of smart meter data may change due to the addition of new electrical equipment or changes in the consumption patterns of residents, that is concept drift occurs [7], which will lead to the degradation or even failure of the prediction performance of existing offline models. Therefore, in the electricity load forecasting task, we need to continuously learn new data generated by the electricity meter without retraining the entire dataset. Therefore, researchers have turned their attention to online learning [8]. This learning mode, which only uses new data to update the model in the process of prediction, greatly reduces the computational burden of the power system and has higher prediction accuracy than traditional offline learning models.

However, most of the current online learning models use the newly arrived electricity load data for regular or quantitative model update [9]. At this time, if the new data and historical data have the same or similar data patterns, there will be invalid updates, which does not only affect the computing resources. Real-time response puts forward higher requirements and has little effect on the improvement of prediction accuracy. Therefore, it is necessary to design a more reasonable online prediction model based on the actual changes of the data.

Therefore, in this paper, an incremental ensemble short-term electricity load prediction model based on sample domain adaptation is proposed, which effectively addresses the above problem using two-stage concept drift detection [10,11] and transfer learning based on sample domain adaptation. In summary, the main contributions of this paper are as follows:

- We propose to combine the significance of change in the distribution of current electricity load data and change in model prediction performance to detect concept drift.
- We design the cumulative-weighted-sampling-based Tradaboost (AWS-Tradaboost) for building the new base model, which solves the problem of inadequate training of the model due to insufficient concept drift samples.
- We develop a novel strategy for updating the weights of the ensemble model.

The rest of this paper is organized as follows. Section 2 is an overview of related work on short-term electricity load prediction. Section 3 describes the proposed AWS-DAIE in detail. Section 4 presents the experiments and corresponding results. Conclusions are presented in Section 5.

## 2. Related Work

Researchers at home and abroad have proposed a large number of short-term load forecasting methods and theories. The existing research is mainly divided into two categories: traditional forecasting methods and artificial intelligence forecasting methods.

Traditional short-term electricity load forecasting methods include the grey model method, fuzzy forecasting method, time series analysis method, and so on. Zhao et al. [12] proposed to optimize the parameters of the grey model GM(1,1) by using the ant colony optimization algorithm and introduced a rolling mechanism to improve the accuracy of electricity load forecasting. Mamlook et al. [13] explored the effect of different parameters including weather, time, and historical data with random perturbations on load forecasting using the priority and importance of fuzzy sets and used fuzzy control logic to reduce forecasting errors. Common time series analysis methods include autoregressive moving average (ARMA), differential autoregressive moving average (ARIMA), generalized autoregressive conditional heteroskedasticity (GARCH), and so on. Reference [14] argued that

there are different levels of noise disturbances in ARIMA forecasting short-term electricity loads, which require re-identification of the model before estimating parameters for the forecasting task, and that the model is able to determine the limit level of noise that the model can tolerate before crashing.

Artificial intelligence methods have emerged in recent years and have been widely used in short-term electricity load forecasting in power systems due to their powerful ability to model complex relationships and adaptive self-learning capabilities. Typical ones include support vector regression (SVR), long short-term memory networks (LSTM), gated recurrent units (GRU), time series convolutional networks (TCNs), and so on. Han et al. [15] extracted meteorological features affecting wind and PV power generation using nonlinear effects and trend correlation measures of the copula function and modeled wind and PV power generation based on LSTM, which is capable of medium- and long-term wind/PV power generation forecasting using limited data samples. Jung et al. [16] used the attention-based gated recurrent unit (Attention-GRU) to model electrical loads in order to tap more key variables in short-term load forecasting tasks and experimentally demonstrated that the prediction performance of the model can be significantly improved when the inputs are long sequences. Gong et al. [17] determined the order sequence of the model in the electricity load by periodically analyzing the correlation of the electricity load data and the fluctuation characteristics of the customer electricity load data. The Seq2seq model is adjusted by combining the residual mechanism (Residual) and two attention mechanisms (Attention) to achieve better results in predicting the actual electricity load data in a certain place.

The combination of models is a new trend in the field of electricity load forecasting, and common combinations include stacking of models and ensemble learning. The stacking of models can fully utilize the advantages of each model to improve the accuracy of load forecasting. Guo et al. [18] constructed historical electricity loads, real-time electricity prices, and weather as the inputs of the model in the form of continuous feature maps. CNN was used to cascade shallower and deeper features at four different scales to fully explore the potential relationships between continuous and discontinuous data in the feature maps. The feature vectors at different scales were fused as the inputs of the LSTM network, and the LSTM neural network was used for short-term electricity load forecasting. Ensemble learning is a new paradigm that combines multiple models to improve the prediction results and can obtain better prediction results than an individual model. Wang et al. [19] used clustering to divide historical electricity load data into multiple partitions, trained multiple LSTM models for different partitions, and finally, used FCC models to achieve the fusion of multiple LSTMs. In this work, the authors used the improved Levenberg–Marquardt (LM) algorithm to train the FCC models to achieve fast and stable convergence. Electricity load forecasting based on decomposition preprocessing has been a hot topic in recent years. Khairalla et al. [20] proposed a flexible heterogeneous integration model that integrates support vector regression (SVR), back propagation neural network (BPNN), and linear regression (LR) learners. The integration model consists of four phases: generation, pruning, integration, and prediction. Li et al. [21] used ICEEMDAN to decompose complex raw electricity load data into simple components and aggregated the final prediction results after forecasting each component individually using a multi-kernel extreme learning machine (MKELM) [22] optimized by grey wolf optimization (GWO). Lv et al. [23] proposed a hybrid model with the elimination of seasonal factors and error correction based on variational modal decomposition (VMD) and long short-term memory networks in response to the dramatic changes that occur in short-term electricity loads due to various factors, which further complicate the data.

Tang et al. [24] used the K-means algorithm to cluster the electricity load, grouped similar data into the same cluster, decomposed the electricity load into several components using ensemble empirical modal decomposition (EEMD), and selected candidate features by calculating Pearson correlation coefficients to construct the prediction input. This

paper selected the deep belief network (DBN) and bidirectional recurrent neural network (Bi-RNN) as the prediction models.

The models mentioned above all belong to the category of offline learning, and in order to learn from new data, researchers have started to research networks and frameworks based on online learning and incremental learning. Von Krannichfeldt et al. [25] advocated combining batch processing and online learning to provide a novel online ensemble learning approach for electricity load forecasting by implementing an improved passive aggressive regression (PAR) model to integrate online into the forecasting results of the batch model to ensure the adaptability of online applications. Álvarez et al. [26] developed online learning techniques for APLF for recursive updating of hidden Markov model parameters and then used the Markov model to model and quantify the uncertainty in forecasting of electricity load data. However, the regular or quantitative update model does not fully consider the actual change of data and requires high real-time response for computing resources, and when the electricity load is in a relatively stationary state, there may be a large number of invalid updates, while also causing a large amount of valuable computing resources to be wasted.

In this paper, we performed concept drift detection in terms of data blocks and adapted to concept drift through domain incremental learning and dynamic adjustment of model weights. When different batch data no longer conform to the static homogeneous distribution assumption, domain incremental learning is able to avoid catastrophic forgetting of historical knowledge while updating the model using only newly arrived data [27]. Concept drift describes the unforeseen shifts in the underlying distribution of streaming data over time [28]. Concept drift detection techniques have emerged in recent years [29–31]; some of them detect concept drift based on the error rate of the model; some detect concept drift using changes in the data distribution; some detect concept drift based on hypothesis testing. To solve the problem that the model cannot be adequately trained due to few concept drift samples, this paper trained the base model by improved transfer learning based on sample domain adaptation [32], which makes the distribution of drift samples approximate the data distribution of that historical data block by adjusting the data weights of the historical and current data blocks and generates the base model based on concept drift samples in the process of adjusting the weights.

## 3. Research Methodology

In this paper, an incremental learning short-term electricity load forecasting model based on sample domain adaptation is proposed. First, two-stage concept drift detection (DSCD) was performed on the current mini-batch samples, and then, we trained a new base model using the sample domain adaptive transfer based on cumulative weighted sampling Tradaboost (AWS-Tradaboost) to solve the problem that the model cannot be fully trained due to few concept drift samples. Finally, we propose a novel incremental ensemble weight updating strategy to construct the final short-term electricity load forecasting model. The processing of the model is shown in Figure 1. The dataset used in this paper was the PRECON dataset [33], and the detailed description of the dataset will be given in the Research Results.

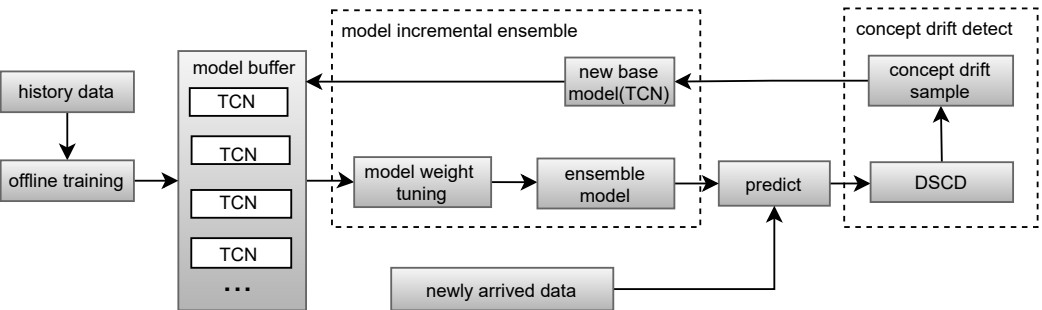

**Figure 1.** Overall process of AWS-DAIE.

### 3.1. Two-Stage Concept Drift Detection

The AWS-DAIE algorithm proposed in this paper only performs incremental updates for concept drift samples, so we need to introduce the two-stage concept drift detection algorithm (DSCD) that we designed. As the name implies, the DSCD algorithm detects concept drift in two stages. The first stage of concept drift detection monitors changes in model performance to determine the degree of adaptation of the current model to new arrivals, while the second stage determines whether concept drift has occurred by checking whether there is a significant difference between the distribution of the current data block and the historical data.

In the first stage of concept drift detection, for each current sample $i$, its corresponding sliding window $W_i$ contains the samples from $i - t$ to $i$. Calculate and compare the absolute value of the prediction error $AE_{i-t}$ of $W_{i-t}$ and the absolute value of the prediction error $AE_i$ of $W_i$ (record the current window and the last full window, respectively); if the absolute value of the prediction error of the window corresponding to sample $i$, $AE_i$, is greater than the absolute value of the prediction error of the window corresponding to a time sample $i - t$, $AE_{i-t}$, $(AE_t > a * AE_{t-1})$, then the current state will be set to drift warning and start collecting the current samples using the adaptive window $W_{adaptive}$. If the data in the adaptive window $W_{adaptive}$ do not reach the preset data amount, the absolute value of the error corresponding to the window corresponding to $m$ consecutive samples is less than $a$ times the absolute value of the prediction error corresponding to the first sample of the adaptive window, and the drift warning state is released. Clear the data in the adaptive window, and start the next round of concept drift detection. If the amount of data in the adaptive window reaches a preset size, start the second stage of concept drift detection. In this paper, we refer to the literature [11] to determine the parameters and select the set of parameter values that make the smallest prediction variance based on a large number of experiments as follows: $a = 1.08$, $m = 5$, $n = 40$.

We collected a certain amount of data under the drift warning state after the first stage of detection, but there may be cases where the data are flagged as a concept drift warning due to noise-induced fluctuations or the model itself is not sufficiently trained. Then, we need to check whether these data are significantly different in distribution from the historical electricity load data, which is the second stage of the concept drift detection. We need to introduce a theorem before introducing the second stage of concept drift detection, namely the paired $t$-test: we let $S$ and $T$ be two independent time series sample, let $S = (S_1, S_2, S_3, \dots S_{n1}) \sim N(\mu_1, \sigma^2)$ and $T = (T_1, T_2, T_3, \dots T_{n1}) \sim N(\mu_2, \sigma^2)$ denote their sample means values, respectively, and let $\sigma_S^2$ and $\sigma_T^2$ denote the variances of samples S and T, respectively, given the original hypothesis of $H0 : \mu_1 - \mu_2 \leq \delta$ at the confidence level $\alpha$, when the statistic t follows the following distribution.

$$t = \frac{S_a - T_a - (\mu_1 - \mu_2 - \delta)}{\sigma_w \sqrt{\frac{1}{n_1} + \frac{1}{n_2}}} \sim t(n_1 + n_2 - 2) \tag{1}$$

$$\sigma_w^2 = \frac{(n_1 - 1)\sigma_S^2 + (n_2 - 1)\sigma_T^2}{n_1 + n_2 - 2} \tag{2}$$

Among them, $n_1 + n_2 - 2$ denotes the degrees of freedom of the distribution. The statistic $t$ lies in the rejection domain at the confidence level $\alpha$ when $t > t_\alpha(n_1 + n_2 - 2)$, at which point, the original hypothesis $H0$ is rejected and $\mu_1 - \mu_2 > \delta$ accepted; at this time, the two samples $S$ and $T$ are judged to be significantly different at the confidence level $\alpha$; the above theory is the criterion for the second stage of concept drift detection. Firstly, historical electricity load data are divided into m blocks of equal size, so that the training data are represented as $T = T_1 \bigcup T_2 \dots \bigcup T_m$, and each block is used to train the base model. Then, we performed down-sampling on these data blocks to obtain $T^{downs}$; the size is denote as *Tsize*. Secondly, the paired $t$-test was performed on the data in the adaptive window $W_{adaptive}$ and $T^{downs}$. When the results of the paired $t$-test between the

data in the adaptive window $W_{adaptive}$ and $T^{downs}$ at confidence level $\alpha$ fall into the rejection domain, the current data block is considered to have concept drift. On the contrary, it is considered to be a pseudo-concept drift caused by noise fluctuation or insufficient training of the model. Then, the data in the adaptive window are used to fine-tune the base model with worst prediction performance among ensemble learning, and clear the data in the adaptive window $W_{adaptive}$, and continue to monitor the changes of the model prediction performance. Through the above two stages, the concept drift samples is used to train a new base predictor when the true concept drift is detected and is used in the subsequent incremental learning process. The overall process of the two-stage concept drift detection algorithm is shown in Algorithm 1.

---

**Algorithm 1** DSCD.

---

**Input:** $D$: newly arrived data; $t$: detecting window size; $a$: concept drift warning threshold; $m$: threshold number of released concept drift warning; $W_a daptive$: adaptive window; $n$: adaptive window size; $T^{downs}$: downsampling of historical data
**Output:** concept drift sample $D_{drift}$

 1: $W_{adaptive}$ = NULL;
 2: *state* = None; count = 0;
 3: number = 0;
 4: **for** sample $d_i \in D$ **do**
 5:     $W_i$: sliding window with the last t samples of $d_i$;
 6:     $AE_i$ = AbsoluteError($W_i$);
 7:     $AE_{i-t}$ = AbsoluteError($W_{i-t}$);
 8:     **if** state==None and $AE_t > a * AE_{t-1}$ **then**
 9:         $W_{adaptive} = W_{adaptive} \cup d_i$;
10:         state = warning;
11:         count = count + 1
12:     **else**
13:         break;
14:     **end if**
15:     **if** state==warning **then**
16:         **if** count < n **then**
17:             **if** $AE_i > a * AE_{i-t}$ **then**
18:                 count = count + 1;
19:             **else** number==m
20:                 state = NULL;
21:                 number = 0;
22:                 number = number + 1
23:             **end if**
24:         **end if**
25:     **else**
26:         state = driftwarning
27:     **end if**
28:     **if** state = driftwarning **then**
29:         Compute sample mean and variance of $W_{adaptive}$ and $T^{down}$ denote as $\mu_{\text{wadaptive}}$,
30:         $\mu_{\text{tdown}}$ and $\sigma^2_{wadaptive}$, $\sigma^2_{tdown}$;
31:         compute double-sided statistics $t_1$;
32:         **if** $t_1 \geq t_\alpha(2n - 2)$ **then**
33:             return concept drift sample $W_{adaptive}$;
34:             state = None;
35:         **else**
36:             state = None;
37:         **end if**
38:     **end if**
39: **end for**

---

*3.2. AWS-Tradaboost*

After detecting concept drift in the previous section, we need to use the concept drift samples to train a new base model. However, the based model may not be adequately trained due to few concept drift samples. This paper used sample domain transfer learning to solve this problem. Tradaboost is a classical sample domain transfer learning algorithm, which continuously reduces the impact of bad data by iterating data weights. During iteration, when the prediction error of a sample in the source domain is large, the influence of the sample on the new base model in the next iteration is weakened by reducing the weight of the sample. It is considered that the training of a sample is insufficient when the model has a large prediction error for this sample in the target domain, and the weight of this sample needs to be increased to better train this sample in the next iteration. The model can better fit the target domain data, and the samples in the source domain data that are closer to the target domain distribution will obtain higher weights through several iteration. However, this algorithm only updates the sample weights according to the current iteration, ignoring the impact of historical iterations on the construction of the current base model. In this paper, we propose an accumulation weight sampling (AWS) method to select the samples with the largest cumulative contribution to the base model training during the iteration process for the next iteration. The current iteration is denoted as $c$, and we need a two-dimensional list of the weights $W$ of each sample in the historical iterations, so that $W$ is denoted as $W = [[w_1^1, w_1^2, \ldots, w_1^c], \ldots, [w_m^1, w_m^2, \ldots, w_m^c]]$, where m is the number of samples in the source domain. The algorithm is described in Algorithm 2.

---

**Algorithm 2** AWS.

---

**Input:** weight vector $W$, source domain sample (*Tsource*)
**Output:** the specified number of samples with the largest cumulative contribution $D^{'}$
  1: Step 1: Calculation of cumulative weighted contribution:
  2: $[awc = [w_1^{ac}, w_2^{ac}, \ldots, w_m^{ac}]$
  3: where is expressed as follows:
  4: $w_i^{ac} = \sum\limits_{j=1}^{c} \lambda^{c-j} w_i^j$
  5: Forgetting factor: The further away from the current
  6: iteration, the smaller the contribution to the current base
  7: model
  8: Step 2: The $k$ samples with the largest cumulative
  9: contribution are selected and recorded as $D^{'}$

---

We need to calculate the similarity between the current drift sample and each historical data block using dynamic time warping (DTW), and the index corresponding to the historical data block with the highest similarity is recorded as $index = \arg \min_{i=1}^{m}(DTW(D_i, D_{current}))$, then return the historical data block *Dindex* corresponding to the index and the base model *Mindex*. We took this historical data block as the source domain dataset Tsource = Dindex; the sample size is denoted as $S$; the concept drift sample was taken as the target domain dataset *Ttarget*; the data size is $T$. In Algorithm 3, we first complete a series of initialization operations including: merging the source and target data to form a new training dataset, setting the maximum number of iterations N, and initializing the weights of the merged dataset $w^1$ and the weights of the learning machine $\rho$. Then, the samples with the largest cumulative contribution to the base predictor training are selected using the AWS algorithm for building the training set of the predictor. Then, calculate the prediction errors of the trained model in the source and target domains, and finally, the weights of the source and target domains are updated in each round of iteration. If the prediction error of the sample in the source domain is larger, it means that the sample is less relevant to the target domain, and the weight of the sample needs to be reduced to reduce its influence. Conversely, if the sample in the target domain has a large prediction error, we need to reduce its weight to improve his impact in the next round, and the final predictor is returned when the number of iterations

reaches the maximum $N$. Therefore, the specific steps of Tradaboost based on cumulative weighted sampling are shown in Algorithm 3.

---

**Algorithm 3** AWS-Tradaboost.

---

**Input:** *Tsource*: original domain sample; *Ttarget*: original domain sample; *Mindex*: the index of Tsource; $N$: the maximum number of iterations;
**Output:** new base model *Mnew*
  1: Initialize: The initial weight vector:
  2: $w^1 = (w_1^1, w_2^1, \dots w_{S+T}^1)$
  3: where $w_i^1 = \{ \begin{smallmatrix} 1/T, i=1,2,\dots T \\ 1/S, i=T+1,\dots S+T \end{smallmatrix}$ $w^{ac} = [[w_1^1], \dots, [w_T^1]]$
  4: $\rho = 1/(1 + \sqrt{2 \ln N})$
  5: **for** $t = 1, 2, \dots, N$ **do**
  6:      $D = AWS(Tsource, w^{ac}) + Ttarget$
  7:      $M_{index} = train(M_{index}, D)$
  8:      calculate error on *Tsource*, *Ttarget*:
  9:      $\varepsilon = \sum\limits_{i=1}^{S+T} w_i^t * \left( \frac{abs(y - \widehat{y_i^t})}{\max(abs(y - y^t))} \right)^2$
  10:      $\beta_t = \varepsilon/(1 - \varepsilon)$
  11:      **if** $i < S$ **then**
  12:         $w_i^{t+1} = w_i^t \cdot \rho^{abs(y_i - \widehat{y_i^t})}$
  13:      **else**
  14:         $W_i^{t+1} = w_i^t \cdot \beta_t{}^{abs(y_i - \widehat{y_i^t})}$
  15:      **end if**
  16:      update cumulative weight:
  17:      $w^{ac} = [[w_1^1, \dots w_1^i], \dots, [w_T^1, \dots, w_T^i]]$
  18: **end for**

---

*3.3. Ensemble Incremental Models*

In this paper, a model buffer was designed to store all the base models corresponding to the current prediction task, and we chose the temporal convolutional network (TCN) as the base model.

The weights of each base model are the most critical factor affecting the prediction performance of ensemble model. Hence, we designed a novel weight update strategy in this paper, which enables the model to adapt to the current electricity load data. We describe our model weight update scheme in detail.

When concept drift occurred, we collected concept drift samples and trained a new base predictor using the AWS-Tradaboost algorithm in Section 3.2 and then added the newly trained base predictor to the model buffer pool. The following scheme was designed to update the model weights of each base predictor:

(1) Take out the conceptual drift sample of the adaptive window in Section 3.1 denoted as $D = (D_1, D_2, \dots, D_{d_t})$; $d_t$ denotes the sample size of the drift sample.

(2) Update and normalize the data weights of the conceptually drifted samples. Predict $D$ using the ensemble model prior to the addition of the new base predictor while calculating the relative error of the prediction results, denoted as $E = (E_1, E_2, \dots, E_{d^t})$, where $E_i$ is denoted as follows:

$$E_i = |H^{t-1}(xi) - y_i^t|)/(\max(H^{t-1} - y^t) \tag{3}$$

where $H^{t-1}(xi)$ represents the predicted value of the ensemble model before adding a new base predictor. $t$ represents the number of base models of the ensemble model.

We first calculate the weights of the concept drift samples by using the following equation.

$$W_i^t = (1/d^t) \cdot er_i{}^{(1-er_i))}, i = 1, 2, \dots, d^t \tag{4}$$

$d^t$ denotes the sample size of the concept drift sample. By endowing the concept drift sample, data weights are used to balance the prediction error of the ensemble model after adding the new base predictor. The data weights are normalized:

$$W_i^t = \frac{W_i^t}{\sum\limits_{j=0}^{t-i} W_i^{t-j}} \tag{5}$$

(3) Construct new concept drift samples $D_{new}$ by assigning the above weight.

$$D_{new} = (D_1 * W_1^t, D_2 * W_2^t, \ldots, D_{d_t} * W_{d_t}^t) \tag{6}$$

(4) Evaluate the prediction error of all base predictors $h_k (k = 1, 2, \ldots, t)$ in the model buffer pool on $D_{new}$:

$$ER_i^t = abs(h_k(x_i) - y_i^t)/(\max(abs(h_k - y^t)) \tag{7}$$

(5) Then, the regression error rate and the regularization error rate are calculated as follows.

$$e_k = \sum_{i=1}^{d^t} W_i^t \cdot ER_i \tag{8}$$

$$\beta_k = \frac{e_k}{1 - e_k} \tag{9}$$

(6) Finally, the weights of each base model are obtained as follows:

$$W_k = \frac{(1/\beta_k)}{(\sum\limits_{k=1}^{t} (1/\beta_k^t))} \tag{10}$$

## 4. Research Results

### 4.1. Dataset Description

This paper conducted an experimental evaluation using the PRECON [33] dataset, which records the electricity consumption patterns of 42 households in Pakistan with varying financial status, daily routes, and load profiles between June 2018 and September 2019. The sampling frequency of the original data is once a minute, and in this paper, we resampled its frequency to once an hour; after resampling, each household contained about 8760 electricity load data. Due to the large number of households involved in the PRECON dataset, it is not possible to clearly and concisely present the algorithm's prediction results on all households; we conducted experiments on the electricity loads of most households and, finally, selected four households (house3, house8, house13, house25) with more significant conceptual drift in the electricity load data and present the algorithm's prediction results on these four households. The ratio of training set and test set was set to 7:3, so the test sets of all four datasets were the electricity load data between 12:00 on 11 February 2019 and 23:30 on 31 May 2019.

We selected the house8 datasets for further analysis. In a non-stationary time series, the distribution and properties of the data change significantly over time, which poses a great challenge and difficulty in the task of short-term electricity load forecasting. This paper used the KPSS test to assess whether the house8 electricity load dataset was stationary. KPSS determines whether a time series is stationary (trend stable) around a defined trend. The original hypothesis was that: the series is trend-stationary. In the KPSS test, if the test statistic is greater than the critical value, the null hypothesis is rejected, indicating that the series is non-stationary.

It can be seen from Table 1 that the test statistic of house8 in the KPSS test was greater than all critical values, which indicates that the electricity load dataset of house8 is non-stationary. This inevitably increases the difficulty of short-term electricity load forecasting.

**Table 1.** KPSS test results of house8.

| Dataset | KPSS | | | | |
|---|---|---|---|---|---|
| | stat | T($\alpha$) | | | |
| | | 1% | 2.5% | 5% | 10% |
| House8 | 2.06 | 0.739 | 0.674 | 0.463 | 0.347 |

Figure 2 shows a part of the test set of house8. The continuous black curve in the figure represents the normally distributed electricity load data; the green square part represents the data segment with concept drift; the blue dot part is the data caused by noise fluctuations.

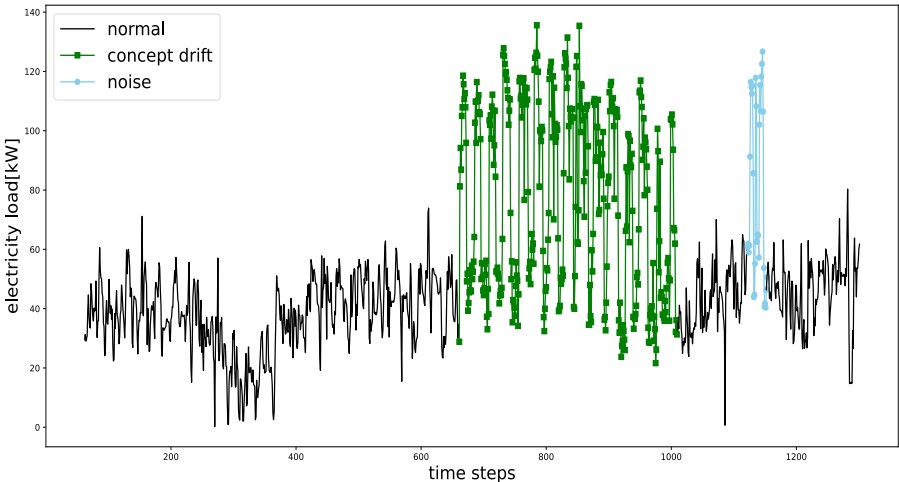

**Figure 2.** Image presentation of house8.

*4.2. Evaluation*

4.2.1. Comparison Algorithms

In order to evaluate the performance of AWS-DAIE, we compared it to some excellent offline training-based and incremental-based classical algorithms in recent years, and the results of the experiments are presented in Section 4.2.3. The compared algorithms included (1) a hybrid of the temporal convolutional network and gated recurrent unit (TCN-GRU) [34], (2) a novel hybrid CNN with LSTM autoencoder-based framework (CNN-LSTM-AE) [35], (3) an incremental ensemble for electricity load model (IncEnsemble) [36], and (4) an incremental ensemble LSTM model (IncLSTM) [37].

Among the above algorithms, TCN-GRU and CNN-LSTM-AE are both traditional offline time series forecasting algorithm. InEnsemble incrementally updates the model using batches of newly incoming electricity load data, that is using quantitative updates to adapt to changes in the data distribution, while IncLSTM is an incremental learning time series forecasting model using transfer learning and ensemble learning, and the base model is a bidirectional LSTM.

4.2.2. Evaluation Metrics

In order to evaluate the prediction performance of the proposed AWS-DAIE and to compare the performance with the above-mentioned algorithms, some metrics need to be determined to measure the performance of each model on the short-term electricity load forecasting task. $X = (X_1, X_2, X_3, \ldots, X_T)$ is the original electricity load observations'

values and predicted values $\widehat{X} = (\widehat{X_1}, \widehat{X_2}, \widehat{X_3}, \ldots, \widehat{X_T})$, where $X$ denotes the average of the electricity load observations values. In this paper, we used the following four metrics: root-mean-squared error (RMSE), mean absolute error (MAE), symmetric mean absolute percentage error (SMAPE), and $R^2$, which are represented as shown below:

$$MAE = \frac{1}{m} \sum_{i=1}^{m} |X_i - \widehat{X_i}| \tag{11}$$

$$RMSE = \sqrt{\frac{1}{m} \sum_{i=1}^{m} (X_i - \widehat{X_i})^2} \tag{12}$$

$$SMAPE = \frac{1}{m} \sum_{i=1}^{m} \frac{|X_i - \widehat{X_i}|}{|(|X_i| + |\widehat{X_i}|)/2|} \tag{13}$$

$$R^2 = 1 - \frac{\sum\limits_{i=1}^{m} (X_i - \widehat{X_i})^2}{\sum\limits_{i=1}^{m} (X_i - X^2)^2} \tag{14}$$

where $m$ denotes the length of the test set. In general, the lower the first three metrics and the higher the last one, the higher the prediction accuracy of this algorithm is. Meanwhile, in order to reduce the influence of random factors in the experiment on the performance evaluation of the algorithm, 10 independent repetitions of each algorithm were conducted on all datasets, and the average result of the 10 experimental results was taken as the final result.

4.2.3. Experimental Results

This section presents the experimental results of AWS-DAIE and the comparison of AWS-DAIE algorithm and several other excellent algorithms on the datasets mentioned above. The experimental setup and the experimental environment should be as similar as possible. The following describes the parameter settings for each algorithm; some uniform settings are described below: the optimizer was Adam, which constructs the input data in a sliding window of window size tw = 6, batchsize = 128, and NVIDIA GeForce GTX 1650Ti was used to accelerate the training of the models in a windows 10 environment. The parameters of each model were set as follows:

a.  TCN-GRU: A TCN model with a kernel size of 3; the number of filters is 64; dilation factors were set to 1, 3, 5, and 7. There were four GRU layers with 64 neurons. The last layer of the GRU was connected to two fully connected layers with 32 and 16 neurons, respectively.
b.  CNN-LSTM-AE: Two Conv1D layers with 64 filters. There were four LSTM layers with 64 neurons, fully connected layers with 32 and 16 neurons, respectively, a maximum pooling layer step size of 2, a a kernel size of 3.
c.  InEnsemble: The base model of ensemble learning model included regression algorithms (MLR, SVR), time series analysis models (AR, Holt–Winters, ARIMA, and so on).
d.  IncLSTM: The number of initial base models was 4; the base model as a fully connected layer with three layers with 64 neurons and two fully connected layers with 32 and 16 neurons, respectively.
e.  AWS-DAIE (proposed): The size of the initial model pool was 4, and the TCN was selected as the base model. The parameters of there TCN were as follows: the kernel size was 3; the dilation factor was set to 1, 3, 5, and 7; the number of filters was 64; the number of neurons was 32 and 16 fully connected layers, respectively.

This section gives the experimental results of AWS-DAIE on four datasets and plots the associated experimental results. In addition, the prediction performance of the AWS-DAIE

algorithm was compared with other excellent algorithms. Figure 3 shows the prediction accuracy metric of several classical time series forecasting algorithms and the proposed AWS-DAIE algorithm on the four households (house3, house8, house13, and house25) of the PRECON dataset. It can be seen from the table that the performance of each algorithm on house3 was relatively the same, while the prediction performance of each algorithm on the house8, house13, and house25 datasets was significantly different. The AWS-DAIE model outperformed the other algorithms in the MAE, RMSE, and $R^2$ on all four datasets, while the SMAPE on the house3 dataset was slightly inferior to the IncLSTM algorithm, but better than the remaining algorithms. Specifically, the MAE of AWS-DAIE on the four datasets was on average 2.40% and 1.16% lower than the two offline algorithms and the two incremental algorithms, respectively. The RMSE on the four datasets was on average 5.10% and 1.60% lower than the two offline algorithms and the two incremental algorithms, respectively. The SMAPE of AWS-DAIE on the rest of the datasets except house3 was on average 4.55% and 2.48% lower than the two offline algorithms and the two incremental algorithms. SMAPE on house3 was 0.03% higher than IncLSTM, but lower than the other algorithms. Finally, on the $R^2$ metric, AWS-DAIE improved the $R^2$ on the four datasets by an average of 11.63% and 4.55% over the two offline algorithms and the two incremental algorithms, respectively.

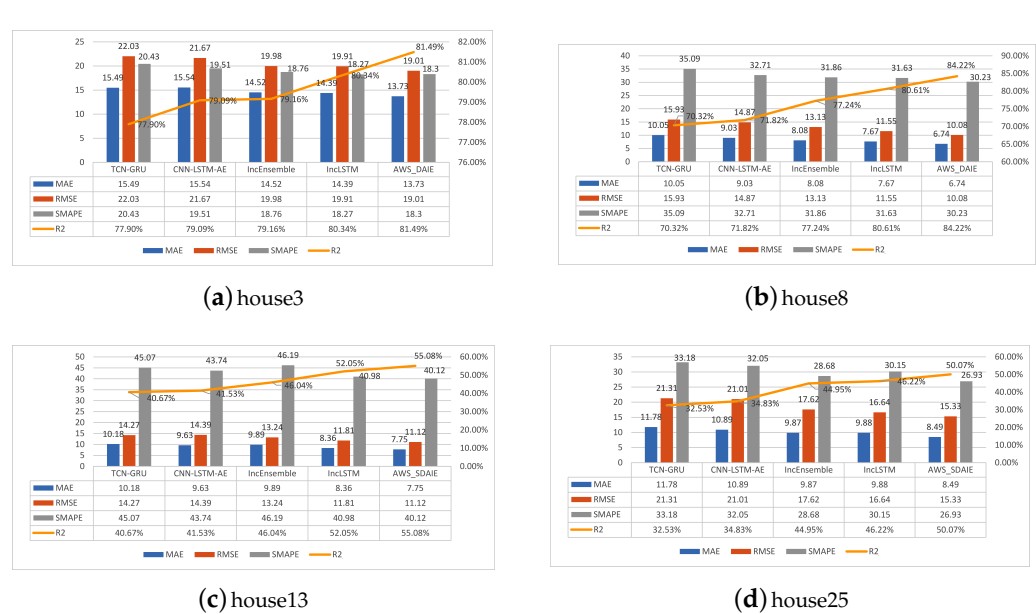

**Figure 3.** Assessment metrics' result graphs.

Although the AWS-DAIE algorithm had a slightly higher SMAPE on house3 than the IncLSTM algorithm, on other metrics, the AWS- DAIE algorithm significantly outperformed the other algorithms on all other metrics, and overall, the AWS-DAIE achieved the most excellent prediction accuracy. As can be seen from these figures above, the prediction accuracy of the incremental-learning-based short-term electricity load forecasting algorithms were significantly better than the offline-learning-based short-term electricity load forecasting algorithms regardless of the dataset. This is because the offline model only uses historical data to build a prediction model and ignores the new information brought by new data, which makes the model unable to adapt to the concept drift existing in non-stationary electricity load data, which also shows that, when predicting non-stationary time series, it is necessary to update the model when the model does not adapt to the current environment. The proposed AWS-DAIE model not only had a significantly better prediction accuracy than the offline-learning-based short-term electricity load forecasting model, but also achieved a higher prediction accuracy than the other two incremental-learning-based short-term electricity load forecasting models; this was due to the fact that the proposed AWS-DAIE

avoids some invalid updates in the process of making short-term electricity load forecasts, and the incremental updates are more in compliance with the changes in electricity load distribution patterns, which also proves that the proposed AWS-DAIE model is able to adapt to conceptual drift in electricity loads in a more timely and accurate manner and can reflect the changes in electricity loads more effectively. The prediction result graph is shown in Figure 4.

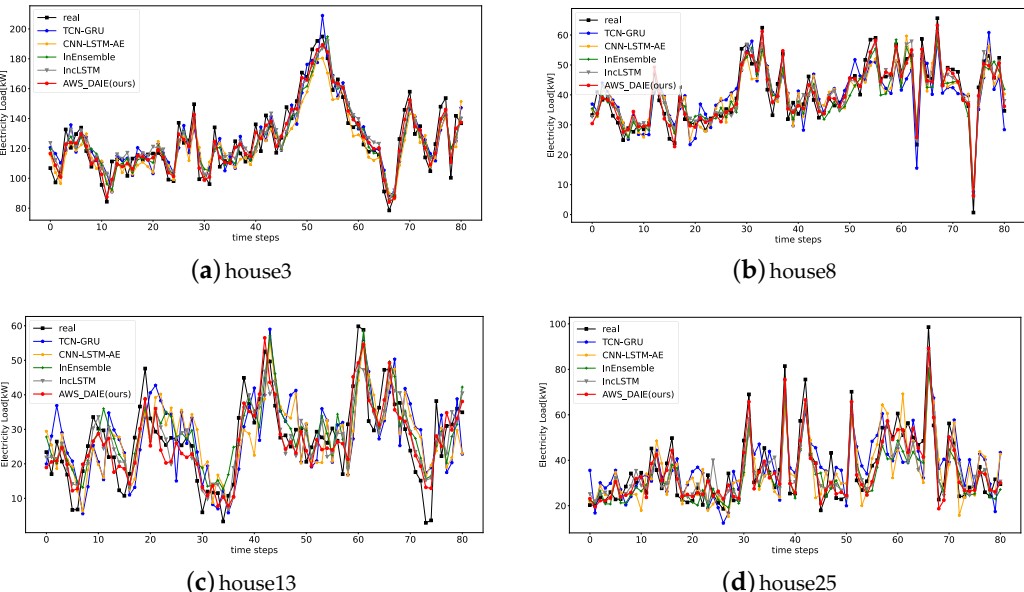

**Figure 4.** Graph of prediction results for each algorithm on four household electricity load datasets.

Figure 5 presents the prediction error absolute value result graphs of AWS-DAIE and other compared algorithms, and Figure 5a–d show the prediction error absolute value curves of each algorithm on house3, house8, house13, and house25, respectively; the curves with red dots in the figure represent the proposed AWS-DAIE algorithm's absolute error graph. It can be seen from the figure that the absolute prediction error curve of the proposed AWS-DAIE algorithm was located below the absolute prediction error curve of the other compared algorithms most of the time, regardless of which dataset it was used on, which means that the prediction value of the proposed AWS-DAIE was closer to the real observed value of the electricity load than the other compared algorithms most of the time. It is not difficult to see that the absolute value curve of the prediction error of AWS-DAIE was relatively less fluctuating compared with the other algorithms, which further indicates that the AWS-DAIE algorithm is more stable and reliable than the other compared algorithms in dealing with the concept drift that occurs in the electricity load. In summary, AWS-DAIE showed better prediction capability compared to both the offline-learning-based algorithms and traditional incremental-learning-based algorithms.

Although the obvious way to compare forecasting performance between algorithms in a time series forecasting task is based on the forecasting performance evaluation metrics described above, this does not allow determining whether the differences between model forecasting performance are significant, which requires the use of other methods, and the Diebold–Mariano test was chosen for testing in this paper. The Diebold–Mariano test is essentially a *t*-test used to assess the relative performance between models. It has been modified recently, so that the test statistic is based on a single time series of error differences, $d_{12}$, defined as follows:

$$DM_{12} = \overline{d_{12}}/\sigma_{d_{12}} \tag{15}$$

$$d_{12} = \frac{1}{N}\sum_{i=1}^{N}\left((e_1)^2 - (e_2)^2\right) \tag{16}$$

where $N$ denotes the sample size, $d_{12}$ is the difference in the mean-squared error of the two models, $\overline{d_{12}}$ and $\sigma_{d_{12}}$ are the mean and standard deviation of $d_{12}$, respectively, and $e_1$ and $e_2$ are the prediction errors of the two models compared, respectively. In this paper, we chose the MAPE as the prediction error. In general, a larger $DM_{12}$ means that Model 1 performs worse than Model 2, and a positive statistic means that the column model outperforms the row model.

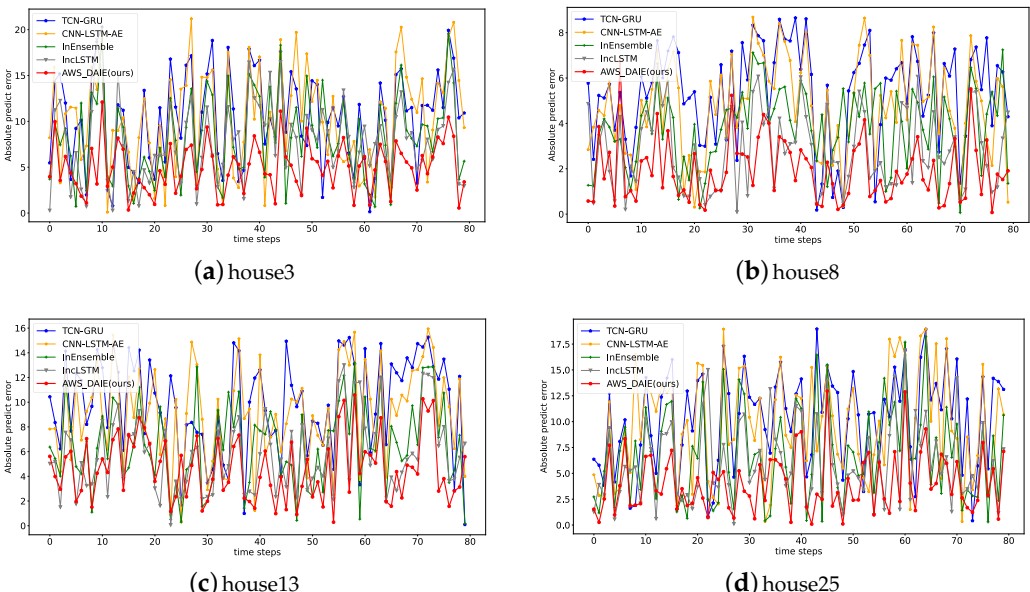

**Figure 5.** Absolute value graph of prediction error for each algorithm on four household electricity load datasets.

As shown in Table 2, the proposed AWS-DAIE model and the rest of the comparison models were performed in this experiment with the Diebold–Mariano test on all electricity load data, and it can be seen that $DM_{12}$ was positive on all four datasets, which means that the proposed AWS-DAIE outperformed several other algorithms, and the incremental-learning-based model outperformed the offline-learning-based model.

**Table 2.** Diebold–Mariano test.

|  | house3 | house8 | house13 | house25 |
|---|---|---|---|---|
|  | **AWS-DAIE** | **AWS-DAIE** | **AWS-DAIE** | **AWS-DAIE** |
| TCN-GRU | 6.24 | 6.22 | 6.03 | 7.53 |
| CNN-LSTM-AE | 5.66 | 5.97 | 6.19 | 6.71 |
| InEnsemble | 2.98 | 4.20 | 3.32 | 3.83 |
| IncLSTM | 2.97 | 3.17 | 2.97 | 3.25 |

## 5. Conclusions

Deep learning has been widely used in the field of short-term electricity load forecasting, but these batch offline models cannot accommodate the concept drift that exists in electricity load data, while the prediction accuracy of the models is subsequently reduced. Regular quantitative update models can adapt to the concept drift to some extent, but there are a large number of invalid updates, which cannot meet the power system's real-time response needs.

The incrementally ensemble short-term electricity load forecasting model based on sample domain adaptation proposed in this paper can effectively solve the above problems. The model is able to update the ensemble model incrementally only after detecting the concept drift of the current electricity load data. Meanwhile, to address the problem that the

base predictor cannot be adequately trained due to the few concept drift samples, this paper fully considered the contribution of historical iterations to the construction of the current base predictor and designed a Tradaboost based on cumulative weighted sampling to better construct the new base predictor. The electricity loads of four households from the PRECON dataset were evaluated, and the proposed algorithm achieved higher prediction accuracy than some current classical offline models, online models, and incremental learning models, which can effectively capture the trend of electricity load and better meet the needs of electricity power systems.

Our research considered concept drift in electricity load forecasting, but did not quantify the extent to which concept drift affects the prediction results of the model. In future work, we will deeply explore and quantify the impact of concept drift on model prediction.

**Author Contributions:** S.L. had the original idea and wrote the paper. J.L. provided review and editing. Y.Z. provided Supervision and professional guidance. All authors have read and agreed to the published version of the manuscript.

**Funding:** Fujian Provincial Natural Science Foundation (2021J01124, 2021J01461).

**Institutional Review Board Statement:** Not applicable.

**Informed Consent Statement:** Not applicable.

**Data Availability Statement:** Data are available in a publicly accessible repository.

**Acknowledgments:** This research was partially supported by the Fujian Provincial Natural Science Foundation (2021J01124, 2021J01461).

**Conflicts of Interest:** The authors declare no conflict of interest.

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
