# Peer review of "AWS-DAIE: Incremental Ensemble Short-Term Electricity Load Forecasting Based on Sample Domain Adaptation"

_sustainability, doi:10.3390/su142114205_

Round 1

Reviewer 1 Report

Considering the variation of customer power consumption mode, this paper proposed incremental ensemble short-term electricity load forecasting based on sample domain adaptation. Firstly, the two-stages concept drift detection method is used to detect the samples with large deviation of power consumption characteristic. Secondly, a new prediction model is established, which can adjust the most suitable training set. Thirdly, the new prediction model with adjustable weights can be integrated into the incremental ensemble model. The detailed comments are as follows,

(1)   A literature review is described in both sections of 0 and 1, and it is recommended to reorganized this part.

(1.1) Furthermore, the ensemble prediction based on several different models is not specifically explained in the section of literature review. The methods of EEMD and VMD are not explained.

(2)   In section 2, for the forecasting method, please give a detailed block diagram, not a flowchart of the algorithm.

(2.1) P5, first paragraph, why select the parameters of a=1.08, m=5, n=40? These parameters did not be explained in the previous description.

(2.2) In section 2.2 and 2.3, the contents are mainly the introduction of the principle, and the specific method description is not clear. How to adjust the weights of the prediction model adaptively? The structure of TCN, including the structure of several TCN integrations proposed in this paper, is not introduced.

(3)   In section 3, only the results of 4 customers are given, which cannot verify the effectiveness of the proposed method. It is recommended to predict all customers in the data set and give the distribution of prediction errors.

(3.1) In Figure 4, please denote what variable the abscissa represents. In addition, the font size of the coordinates is too small to be clearly seen.

Author Response

Response to Reviewer 1 Comments

Point 1: A literature review is described in both sections of 0 and 1, and it is recommended to reorganized this part.

Response 1: The introduction and related work of the article have been reorganized, with the related work focusing on the literature review and the introduction focusing on the introduction of the problem, where the detail modifications are highlighted in red.

Point 2: Furthermore, the ensemble prediction based on several different models is not specifically explained in the section of literature review. The methods of EEMD and VMD are not explained.

Response 2: Ensemble prediction based on several different models have been supplemented and the methods of EEMD and VMD have been specifically explained, which makes the literature review complete and sufficient, specific details of the modifications are highlighted in the related work.

Point 3:  In section 2, for the forecasting method, please give a detailed block diagram, not a flowchart of the algorithm.

Response 3: The algorithm flowchart of the proposed model has been updated to a detailed algorithm module, which makes the algorithm description much more clear and concise,  and the details of the modifications are highlighted in Section 2.1.

Point 4: P5, first paragraph, why select the parameters of a=1.08, m=5, n=40? These parameters did not be explained in the previous description.

Response 4: With regard to the parameter setting of the concept drift detection, this paper refers to the parameter determination methods of some related literature on concept drift detection based on accuracy thresholds, and selects a set of parameters that minimize the prediction variance based on extensive experiments.

Point 5:  In section 2.2 and 2.3, the contents are mainly the introduction of the principle, and the specific method description is not clear. How to adjust the weights of the prediction model adaptively? The structure of TCN, including the structure of several TCN integrations proposed in this paper, is not introduced.

Response 5: The algorithm descriptions in Sections 2.2 and 2.3 have been reorganized and refined to make them as clear and readable as possible, while the introduction of the TCN algorithm has been removed, and the details of the modifications are highlighted in the returned manuscript.

Point 6:  In section 3, only the results of 4 customers are given, which cannot verify the effectiveness of the proposed method. It is recommended to predict all customers in the data set and give the distribution of prediction errors.

Response 6: Since there are 42 households involved in the PRECON dataset, it is difficult to present the electricity load prediction results of all households in a clear and concise manner. In fact, we have done experiments on almost all households, but only four households with more obvious concept drift phenomenon are selected to elaborate the prediction results, which have a series of concept drift phenomenon that can better validate our proposed algorithm and ensure a concise and intuitive presentation.

Point 7: In Figure 4, please denote what variable the abscissa represents. In addition, the font size of the coordinates is too small to be clearly seen.

Response 7: The abscissa denote time steps, the font size of the coordinates has been resized so that they are clearly displayed. In addition, the image used in this paper is a vector image, which can be enlarged to see the details of the image more clearly, Specific details of the revisions are presented in 3.2.3. Experimental results.

Reviewer 2 Report

The paper has been presented a method for prediction of small loads such as home load profile. This topic is very challenging because of the vague nature of the loads.

1- The improvement in the accuracy of the proposed method is striking. What is the reason of high improvement based on the opinion of the authors?

2- Is this method suitable for prediction of special days load profile?

3- Is the load profile of a home predictable? There are multiple unpredictable parameters which are effective in load demand such as party, football match and etc. What's the authors opinion?

Author Response

Response to Reviewer 2 Comments

Point 1: The improvement in the accuracy of the proposed method is striking. What is the reason of high improvement based on the opinion of the authors?

Response 1:  It is well known that offline modelling of historical electricity load data is based on the assumption of independent homogeneous distribution, but the distribution of future electricity data in non-stationary household electricity loads may have a completely different distribution from that of historical electricity loads, and our model takes this feature into account by modelling patterns of information that may be implied in the newly arrived electricity loads that are not available in the historical data, and by gradually improving the predictive performance of the model through the process of constantly adapting to new patterns. We believe this is the reason for the high prediction accuracy achieved by our algorithm.

Point 2: Is this method suitable for prediction of special days load profile?

Response 2:  This method is suitable for prediction of special days load profile.

(A) If the historical data already contains a pattern of special days load profile, then special days load profile are already taken into account when modelling the historical data offline.

(B) If the historical data does not contain special days load profile, this special day’s load profile will show a different distribution pattern than the historical data.

Our algorithm will detect the presence of this special days load profile during the concept drift detection and model special days load profile in time and adapt well to the special holiday pattern in subsequent forecasts.

Point 3: Is the load profile of a home predictable? There are multiple unpredictable parameters which are effective in load demand such as party, football match and etc. What's the authors opinion?.

Response 3:  Whether it is a home load profile or a factory load profile, and so on, these load profile have a certain periodical pattern of variation most of the time, so they are all predictable. And the multiple unpredictable parameters you mentioned, such as party, football match and etc. These situation may be concept drift detected by our algorithm. Similar to the special days load profile, our algorithm will model these special data patterns to continuously improve the model's ability of predict special abrupt situations and various special holidays, which is the core of our work.

Reviewer 3 Report

1. Significant testing should be considered to evaluate the model performances. For example, the Diebold-Mariano test, or Friedman test + post-hoc Nemenyi test.

2. Performance of the proposed method can be better analyzed by using more visualized graphs.

3. Advantages of proposed model compared with existing offline models and online models should be described more clearly.

4 The model complexity or time consumption can be further presented.

Author Response

Response to Reviewer 3 Comments

Point 1: Significant testing should be considered to evaluate the model performances. For example, the Diebold-Mariano test, or Friedman test + post-hoc Nemenyi test?

Response 1: Relevant content about the Diebold-Mariano test has been supplemented at the end of the experiment in this paper.  Detail revision can be seen in Section 3.2.3.

Point 2:  Performance of the proposed method can be better analyzed by using more visualized graphs.

Response 2: Regarding the presentation of the visual images, some details of the absolute prediction error plots for each algorithm have been optimised and visualised the prediction evaluation metrics section, so that it can illustrate more clearly the comparison between the individual algorithms. Specific modification details are highlighted in Section 3.2.3 Experimental results.

Point 3: Advantages of proposed model compared with existing offline models and online models should be described more clearly.

Response 3: Some of the descriptions in this section have been revised and reorganised, and some test descriptions have been supplemented in order to compare in detail the existing offline algorithms, the online algorithms and the proposed AWS-DAIE. Specific details of the revision are highlighted in section 3.2.3 Experimental results.

Point 4: The model complexity or time consumption can be further presented.

Response 4: Because of the existence of various type of deep learning models, which results in the selection of different base models, it is difficult to give the time complexity of each algorithm.

Reviewer 4 Report

Decision: Reject

Summary

It is important of predicting electricity consumption by machine learning based on real data from households. Especially, the adaptive learning model would be important research in the future.

On the other hand, Figure.3 shows overestimations of the predicted results. “Time” is not described for the results of electricity consumption, which fluctuates with time and season. The results of the predictions by the model for each of the selected households are not clearly show. In addition, machine learning that handles time series is strongly affected by hyperparameters, then these should also be clearly described these. As above, there are many problems on this paper.

At now, I cannot determine if the models in this paper are appropriate or not. I rejected the paper because it is difficult to revise in a short period improving the many figures and manuscripts proofreading changes or makes required.

Author Response

Response to Reviewer 4 Comments

Point 1: On the other hand, Figure.3 shows overestimations of the predicted results. “Time” is not described for the results of electricity consumption, which fluctuates with time and season.

Response 1: Figure.3 in the original manuscript is only a visual illustration of the characteristics of the dataset rather than a presentation of the predicted results, and the axes (time steps, etc.) of the article graph have been improved in the revision.

Point 2: The results of the predictions by the model for each of the selected households are not clearly show. In addition, machine learning that handles time series is strongly affected by hyperparameters, then these should also be clearly described these.

Response 2: The work center of this article not on the quest for high forecasting accuracy of the model by exploring the appropriate hyper-parameters, but on how to update the forecasting model to reduce the computational time and resource wastage caused by regular or quantitative model updates after different distribution patterns of power load data (the concept drift mentioned in the paper), and at the same time achieve a satisfactory prediction result.

The hyper-parameter settings of the model in this paper mainly refer to the comparison algorithms and the setting suggestions of the base model we use, while the threshold parameters of the concept drift detection stage are determined with reference to the setting method of the relevant literature for concept drift detection based on accuracy, which are mentioned in our article.

In the end, we also read our own article several times, and made a lot of revisions to improve the scientificity and readability of the article. The revision details of the article have been highlighted in the revised manuscript.

Round 2

Reviewer 1 Report

This version is modified according to the comments.

Author Response

Thank you for your approval!

Reviewer 4 Report

Decision: Major Revision

Summary

It is important of predicting electricity consumption by machine learning based on real data from households. Especially, the adaptive learning model would be important research in the future.

I think that this version is a significant improvement from the previous one. However, the results of the electricity load forecasting, which is important for this time-series analysis, are not show graphically. Then, it is difficult to determine the results of the time-series analysis only by the error indicators, please provide the forecasting results of the electricity load by your model.

Please add a line number.

Each part

1P

Please add “Sustainability” icon.

3P Line 4 form bottom

Integrated empirical model decomposition (EEMD) -> Integrated empirical model decomposition (IEMD)

Section 3.1

“Dataset” should be described in the methodology part.

Figure.2

Please add legends.

As a result of the forecasting, please show the 4 households analyzed data in the figure that forecasted the "electricity load" as shown in this figure. It is difficult to determine if the model is appropriate as a result of time-series forecasting based on the only error indexes results.

Figure3 & 4

The text in these figures is small and the resolution of the figures is low. Then, it is difficult to read the text in the figures.

Please change the font and resolution to make it easier for readers to read.

Author Response

Response to Reviewer 4 Comments

Point 1: Please add “Sustainability” icon.

Response 1: The "Sustainability" icon has been added to the first page of the article.

Point 2: 3P Line 4 form bottom

Integrated empirical model decomposition (EEMD) -> Integrated empirical model decomposition (IEMD).

Response 2: I apologize for the misspelling of "Ensemblel Model Dcomposition (EEMD)" as "integrated empirical model decomposition (EEMD)", which has been corrected at the bottom of page 3.

Point 3: “Dataset” should be described in the methodology part.

Response 3: Thank you very much for your valuable comments and with reference to other articles in sustainability journal, a brief description of the dataset has been given in the methodology part, which is highlighted in red in page 4, and the specific description of the dataset remains in the Research Results.

Point 4: Please add legends.

As a result of the forecasting, please show the 4 households analyzed data in the figure that forecasted the "electricity load" as shown in this figure. It is difficult to determine if the model is appropriate as a result of time-series forecasting based on the only error indexes results.

Response 4: Plots of the model's predicted results for four household electricity loads have been added to the experimental results section of the article(page 13).

Point 5: The text in these figures is small and the resolution of the figures is low. Then, it is difficult to read the text in the figures.

Please change the font and resolution to make it easier for readers to read.

Response 5: The font of the pictures in the article has been adjusted, in addition, the pictures in this article are vector pictures(.svg), you can clearly see the specific details of the pictures by scrolling the wheel to enlarge.

Round 3

Reviewer 4 Report

Decision: Minor Revision

Summary

I think that this revised version was became the paper worthy of discussion.

Some part would be necessary improve, and I would appreciate it if you could publish it after the improving.

Each part

Figure.2

Please add legends and units even if they are images.

P12 16 Line from bottom

R2score -> R2 score

There are some mistakes in the notation.

Figure.3

Please increase the resolution of this figure.

Figure.4

Please add the unit of Electricity load.

Author Response

Point 1:

Figure.2  Please add legends and units even if they are images.

Figure.3  Please increase the resolution of this figure.

Figure.4  Please add the unit of Electricity load.

P12 16 Line from bottom(R2score -> R2 score)  There are some mistakes in the notation.

Response 1:

A legend and units have been added to Figure 2. In response to the lack of resolution in Figure 3, we have adjusted Figure 3 to improve its resolution for clarity. Units have also been added to the vertical axis of Figure 4. Finally, the spelling errors in the article (e.g. R2score -> R2 score) have been checked and corrected.
